# Virus Infections Play Crucial Roles in the Pathogenesis of Sjögren’s Syndrome

**DOI:** 10.3390/v14071474

**Published:** 2022-07-04

**Authors:** Kunihiro Otsuka, Mami Sato, Takaaki Tsunematsu, Naozumi Ishimaru

**Affiliations:** Department of Oral Molecular Pathology, Tokushima University Graduate School of Biomedical Sciences, Tokushima 770-8504, Japan; ootsuka.kunihiro@tokushima-u.ac.jp (K.O.); c301951007@tokushima-u.ac.jp (M.S.); tsunematsu@tokushima-u.ac.jp (T.T.)

**Keywords:** virus, EB virus, autoimmunity, Sjögren’s syndrome

## Abstract

Sjögren’s syndrome (SS) is an autoimmune disease especially targeting exocrine glands, such as the salivary and lacrimal glands. A radical therapy for SS based on its etiology has not been established because of the complex pathogenesis of the disease. Several studies have demonstrated a relationship between virus infection and SS pathogenesis. In particular, infection with the Epstein-Barr (EB) virus among others is a potent factor associated with the onset or development of SS. Specifically, virus infection in the target organs of SS triggers or promotes autoreactive responses involving the process of autoantigen formation, antigen-presenting function, or T-cell response. Our review of recent research highlights the crucial roles of virus infection in the pathogenesis of SS and discusses the critical association between virus infection and the etiology of autoimmunity in SS.

## 1. Introduction

Sjögren’s syndrome (SS) is an autoimmune disease, in which the autoimmune response targets exocrine glands, such as the salivary and lacrimal glands [1,2]. Dry eye and mouth are the main symptoms, adversely affecting the quality of life for patients with SS [3]. In some patients with SS, glandular lesions are also accompanied by extra-glandular lesions in the lung, kidney, and joint [4,5,6]. Based on the affected organs, the disease is classified to primary SS and secondary SS [7,8]. In primary SS, the disease is limited to the salivary or lacrimal glands. In secondary SS, the primary process is accompanied by other connective tissue diseases, such as rheumatoid arthritis (RA), systemic lupus erythematosus (SLE), and scleroderma [9,10,11,12]. The clinical dynamics and molecular mechanisms underlying the onset and development of SS have proven complex enough to elude a radical therapy for SS.

Autoimmune diseases, including SS, are caused by multiple factors involving genetics, the environment, or a dysfunction of the immune system or the target organs [13]. The resulting breakdown of the peripheral or central immune tolerance occurs as autoimmunity. Immunological homeostasis is maintained by the precise maintenance of all components of the immune system, innate and acquired immunity, or cellular and humoral immunity [14,15].

Infectious diseases, such as virus infections, can induce an imbalance or dysfunction of the immune system [16,17,18]. For instance, the human T-cell leukemia virus type 1 (HTLV-1), the human immunodeficiency virus (HIV), and the Epstein-Barr virus (EBV) directly infect hematopoietic cells, including T cell, B cell, or macrophages, to induce an impairment of the normal immune response, or immunodeficiency [19,20,21,22]. In addition, local inflammation in the target organ of the virus affects systemic immune responses [19,23].

Infections by EBV, HTLV-1, HIV, cytomegalovirus (CMV), and hepatitis viruses have been reported to be associated with the pathogenesis of SS [24]. Salivary gland cells, in particular, have been well known to be targeted by various viruses [24]. Therefore, the unique features of salivary gland cells may explain the link between SS and virus infection. In this review article, we discuss the clinical and pathological aspects of the association of virus infection with the pathogenesis of SS. In addition, we review recent reports on the relationship between the viral immune response via the toll-like receptor (TLR) or type 1 interferon (IFN) signaling and the onset or development of SS.

## 2. Autoimmunity and Viral Infection

A lot of reports have demonstrated that virus infection plays a key role in the pathogenesis of a variety of autoimmune diseases, including SS, RA, SLE, and multiple sclerosis [16,19,23,25,26]. Molecular mimicry, bystander activation, and epitope spreading have been identified among the complex cellular or molecular mechanisms associated with the onset or development of autoimmune diseases induced by virus infection [27,28].

Molecular mimicry, known for its association with various autoimmune diseases [19,23,28], involves a virus protein (antigen) which is structurally similar to a protein of the body (self-antigen), which the B or T cells’ antigen response erroneously also targets, leading to autoimmune response [28]. Based on the feature of protein similarity between viruses and host, four types of molecular mimicry have traditionally been proposed. Type 1 is “complete identity” at the protein level; Type 2 is “homology” at the protein level between virus and host. Type 3 is “common or similar amino acid sequences or epitopes”; Type 4 is “structural similarities” between the virus and host [28]. For instance, several studies of autoimmune diabetes mellitus, known as type 1 diabetes (T1D), have provided evidence that the similarity between viral protein components of the Coxsackievirus and pancreatic β-cell antigen glutamic acid decarboxylase 65 triggers the activation of T cells in the pathogenesis of T1D [29,30].

In the bystander activation model, the antiviral immune response induces localized inflammation with the release of self-antigens in the damaged tissue. These are subsequently processed and presented by antigen-presenting cells, such as dendritic cells or macrophages, to enhance the activation of autoreactive T cells during the process of autoimmunity [27]. By contrast, in epitope spreading, a persistent viral infection is believed to trigger the release of self-antigens, which are not cross-reactive with the virus antigen involved in the original immune response, to activate autoreactive T cells in succession, thereby spreading additional autoreactive epitopes of self-antigens to induce autoimmune response [26]. Infection by various viruses, including the influenza virus, HIV, herpes simplex virus (HSV), Epstein-Barr (EB) virus, Coxsackievirus, human herpesviruses, hepatitis C virus, Sendai virus, cytomegalovirus (CMV), Dengue virus, and Parvovirus, is closely associated with the onset of an autoimmune disease through the molecular mechanisms leading to autoantigen formation [23].

## 3. Sjögren’s Syndrome and Viral Infection

A variety of viruses, including EBV, HTLV, HSV, HBV, HCV, and CMV, have been reported to be associated with the development of SS [24,31], and infection with these viruses is known to be a clinical risk factor [1]. Pathogenesis of SS is very complex, as evidenced by multiple lesions observed in glandular and extra-glandular organs. A typical pathological feature of the salivary glands of patients with SS is a focal accumulation of lymphocytes around the salivary duct (Figure 1). The inflammatory lesions in the target organ are formed during a complicated process via the breakdown of central or peripheral immunological tolerance, affected by genetic and/or environmental factors, including viral infection [3].

During the first step of the peripheral immune response against viral infection, antigen-presenting cells (APCs), such as macrophages and dendritic cells, play a key role in the activation of autoreactive T cells as well as virus-specific T cells or natural killer cells [34]. On the other hand, viral infection in target organs, such as the salivary glands, also play an important role in the formation of autoantigens via molecular mimicry or epitope spreading [3]. Cell death, including apoptosis, of salivary gland cells is known to induce the secretion of inflammatory mediators as well as autoantigen formation (Figure 2). A multiple process, including the peripheral immune response to viral infection and the infectious process in salivary gland cells, is associated with the formation of autoimmune lesions in the target organ (Figure 2). Moreover, in the chronic stage, a variety of immune cells, such as the follicular helper T (Tfh) cells, B cells, and plasma cells are also involved in both antiviral and autoimmune responses, suggesting the activation of humoral immunity related to autoantibody production (Figure 2) [34].

The salivary gland involvement among the target organs of a virus is a crucial point in the pathogenesis of primary SS. For instance, CMVs directly infect salivary gland cells leading to inflammatory lesions similar to SS lesion [29,35]. Furthermore, in CMV-infected salivary glands, one pathological feature is an “intranuclear inclusion body” in the salivary ductal cells (Figure 3). Similarly, lymphocytic infiltration resembling the pathology of SS lesion was observed (Figure 3). However, the antibody level was not significantly increased in patients with SS [36,37]. Moreover, sicca syndrome, characterized by chronic sialadenitis, has been reported in patients infected with HCV [19,38].

A resect report demonstrates that all the HCV patients had low saliva flow, chronic sialadenitis, and lacked anti-SSA or anti-SSB autoantibodies [39]. Moreover, a significant increase in the number of inflammatory cells with predominantly CD8^+^ T cells was observed in the histology of lip biopsy from HCV patients, suggesting that HCV infection may cause xerostomia by the mechanisms distinct from SS [39]. Moreover, HTLV-1 infection in the salivary gland promotes cytokine production, such as TGF-β, to enhance fibrosis in the salivary gland tissues of anti-centromere-antibody-seropositive SS patients [40,41]. The pathogenesis of HTLV-1-related SS has different immunological patterns in contrast to primary SS [42]. Thus, the salivary glands targeted by any virus do not directly affect the onset or development of SS.

After viruses infect salivary gland tissues, plasmacytoid dendritic cells (pDCs) catch viral antigens and, in response, produce a variety of cytokines or chemokines to activate salivary gland epithelial cells through TLRs, such as TLR2, 3, and 7 [24,43]. RNA viruses, such as HCV, are recognized by TLR3, 7, 8, and 9 of antigen-presenting cells [44]. However, DNA viruses, such as EBV, HSV, HPV, and CMV, are recognized by the TLR-independent pathway or TLR-9 [44,45]. Salivary gland cells may function as nonprofessional APCs through both TLR-dependent and -independent pathways. Activated salivary epithelial cells produce or upregulate many immune-related molecules, including type I interferon, BAFF, co-stimulatory molecules, and adhesion molecules [24,43,46]. Moreover, activated salivary epithelial cells can function as a nonprofessional APC, suggesting that the ectopic immune response within the salivary gland tissues may trigger autoimmunity [47,48]. Similarly, increased expression levels of type 1 IFN or type 1 IFN-induced genes have been observed in many patients with primary SS and other autoimmune diseases. Those expression levels correlated with endogenous virus-like genomic repeat elements that contribute to the initiation or amplification of SS [49].

## 4. Epstein-Barr Virus and Sjögren’s Syndrome

### 4.1. EBV and Its Clinical Siginificance

In contrast to various viruses that are merely associated with the pathogenesis of SS, EBV has been repeatedly reported to contribute closely to many processes of the development of SS [24,50,51]. In addition to the role of T-cell hyperactivation discussed above, B cell hyperactivation leading to autoantibody production plays an important role in the formation of autoimmune lesions in SS as well as T cell hyperactivation [3]. Consistent with that, the risk of B cell lymphoma is increased in the patients with SS [52]. Over 90% of the population in the world is widely infected with EBV, which infects immature or naive B cells and survives in mature or memory B cells to continue a persistent infection [25]. Although EBV infection of resting memory B cells is latent, EBV can run repeated cycles of lytic replication during B cell stimulation and plasma cell differentiation [3], indicating that the reactivation of EBV depends on B cell stimulation or differentiation. Consistent with this notion, multiple studies revealed that reactivation of EBV-infected B cells in the patients with SS mainly affected B cell polyclonal activation leading to autoantibody production, including anti-SSA or SSB autoantibody [50,51].

Surprisingly, although it is well known that in patients with SS, EBV DNA is detected at high levels in salivary gland tissues and mononuclear cells in peripheral blood [53], EB nuclear antigen (EBNA) remains hardly detectable for immunohistochemical analysis of the same tissues. Specifically, we could not detect EBNA-positive cells in the biopsy sections of minor salivary gland tissues of not only healthy controls but also patients with SS from Grade 0 to 4 [32,33], including those exhibiting a formation of lymphoid follicles (Figure 4). However, EBNA-positive cells were detectable in the tissue sections of patients with methotrexate-associated lymphoproliferative disorder (MTX-LPD), aggregate term lumping heterogenous clinicopathological entities developing in patients receiving immunosuppression agents, such as MTX, and is known to be closely associated with EBV reactivation (Figure 4) [54]. Together, these findings suggest that EBV continues to replicate inconspicuously, detectable at DNA level, during chronic inflammation in SS, which is quite different from the vigorous EBV replication in MTX-LPD.

### 4.2. Molecular Pathogenesis of EBV-Associated Sjögren’s Syndrome

Investigations of the pathogenic factors active during chronic B cell activation in SS have focused on several molecules, such as B cell activating factor (BAFF), β2 microglobulin (β2M), and serum-free light chains [3,55,56]. High levels of anti-EBV early antigen (EA) antibody, associated with viral reactivation, were observed in the patients with SS who were positive for anti-SSA/B [19]. Moreover, levels of β2M, κ and λ light chains, and immunoglobulin G were observed to be higher in anti-EA positive than anti-EA negative patients with SS [51]. By contrast, there was no difference of EBV DNA levels in the circulating compartment between SS and controls [51], suggesting that EBV reactivation and replication may be efficiently regulated in the peripheral blood in the patients with SS. Thus, despite the important results obtained from studies based on clinical samples, including blood, and biopsy materials, the molecular pathogenesis of EB-associated SS remains unclear.

Among the various environmental risk factors as a risk for the development of autoimmune disease, 2,3,7,8-tetrachlorodibenzo-p-dioxin (TCDD) in particular has been suggested to contribute to the onset of autoimmune diseases, including SS, through the impairment of T-cell differentiation and functions [57]. Furthermore, TCDD was shown to upregulate BZLF1 transcription, leading to EBV replication in EBV-infected B cell lines and a salivary gland cell line via the aryl hydrocarbon receptor (AhR), a ligand-activated transcription factor for TCDD [58]. Interestingly, the transcription activity of BZLF1 and cytochrome P450 1A1 (CYP1A1), another TCDD/AhR-responsive gene, were enhanced in the saliva of patients with SS compared with controls [58]. These experimental results suggest that environmental pollution can present risk factors for the onset or development of autoimmune diseases via EBV reactivation.

A significant path of molecular pathogenesis of SS associated with EBV infection is molecular mimicry of antigens that come into two flavors, (i) SS-A, where the viral EBNA-2 protein mimics the Ro-60 antigen and (ii) SS-B where EB virus-encoded small RNA-1 (EBER-1) and EBER-2 viral proteins mimic the La antigen (SS-B) [59,60]. Initially, the EB virus proteins were identified as target molecules for autoantigens in SLE [59,60]. Although the detailed mechanism of the molecular mimicry in SS has not been fully understood, anti-SS-A and anti-SS-B autoantibodies have proven to be widely useful as biomarkers for the diagnosis of SS.

Recently, transcriptional changes based on comprehensive gene expression were evaluated by single-cell RNA-sequence analysis using PBMCs in healthy controls and patients with primary SS [61]. Several differentially expressed genes were shared by multiple subpopulations of immune cells, such as T and B cells [61]. Furthermore, abnormal signaling pathway, including antigen processing and presentation, the B cell receptor signaling pathway, the T cell receptor (TCR) signaling pathway, and virus-related pathways, such as influenza A, CMV, and EBV, was highly enriched in patients with primary SS [61]. Notably, the sequence among TCR α- and β-chains were associated with highly prevalent virus infection, including EBV [61].

## 5. Animal Models of Virus Infection-Associated SS

Mice with an HTLV-1 tax transgene were reported as the first virus infection-associated models for SS [62]. The proliferation of salivary ductal epithelial cells was observed in the submandibular, parotid, and minor salivary glands in HTLV-1 tax transgenic (TG) mice within several weeks after birth [62]. After around five weeks of age, lymphocytes were detectable adjacent to the proliferation of epithelial cells. As the proliferation progressed, lymphocytes as well as plasma cells infiltrated diffusely in the salivary gland tissue in the TG mice [62]. The HTLV-1 tax gene has been detected in salivary gland tissues among patients with SS [41]. Although the molecular mechanism via the HTLV-1 tax gene has been unclear, morphologic change of salivary ductal cells with cell growth or death may affect the onset of inflammation resembling SS lesion in HTLV-1 tax TG mice.

To determine the significance of apoptosis mediated by Fas and tumor necrosis factor receptor I (TNFRI) in virus infection-induced SS lesions, C57BL/6 (B6) background Fas gene-deficient *lpr*/*lpr* (B6-*lpr*/*lpr*) mice, B6-tnfr knockout (KO) mice, B6-*lpr*/*lpr*-*tnfr*KO mice, and B6-wildtype mice were intraperitoneally infected with mouse CMV (MCMV) [63]. Acute inflammatory lesions were observed in all groups at 28 days after the injection [63]. However, at 100 days after the infection, severe inflammation developed and increased levels of autoantibodies, such as rheumatoid factor, anti-dsDNA, anti-SSA, and anti-SSB antibody, was observed only in the B6-*lpr*/*lpr* mice [63]. Therefore, Fas-mediated apoptosis plays a vital role in the downmodulation of the immune response against virus infection. A defect of Fas-mediated apoptosis may lead to chronic inflammation resembling SS lesions in the postinfection.

In addition, NZB2328 mice, an autoimmune-prone mouse strain, were intraperitoneally infected with MCMV, and the chronic inflammation was observed in the salivary and lacrimal glands 100 days after the infection. In comparison, acute inflammation was detected 28 days after the infection [64]. Notably, severe inflammatory lesions revealing focal mononuclear cell infiltration in the periductal area were found in MCMV-infected female mice, resembling SS lesions [64]. Moreover, autoantibody levels of sera, such as anti-SSA and anti-SSB antibodies in MCMV-infected NZB2328 mice, were significantly higher than those in control mice [64]. The results imply that MCMV infection may induce or enhance peripheral autoimmune response leading to chronic inflammation. Therefore, the animal model using MCMV has been widely used in many studies to understand the pathogenesis of autoimmunity and the molecular or cellular mechanisms of the breakdown of immune tolerance [65].

## 6. Future Perspectives

A recent study demonstrated that a novel severe acute respiratory syndrome coronavirus (SARS-CoV-2) infects salivary gland cells by anchoring its spike glycoprotein to ACE2, one of the surface receptors of the host cell [66]. In addition, viral particles, proteins, nuclear acid, and antivirus immunoglobins are detectable in the saliva from patients infected with SARS-CoV-2 [67]. This suggests that since exocrine glands, such as the salivary gland, are placed in the front line of our immune system to protect the body through secretion, a virus-infected salivary gland may become a target organ of autoimmune responses during the complicated process of viral infection.

The pathogenesis of infectious disease by SARS-CoV-2 has been gradually unraveled. The effect or impact of SARS-CoV-2 infection on patients with patients with autoimmune diseases, including primary SS, has also been reported in the point of general emotional status, or various symptoms of SS [68,69]. Dry mouth due to SARS-CoV-2 infection is well known, and it has been suggested that the dry mouth of patients with SARS-CoV-2 infection may be induced by the neuroinvasive and neurotrophic properties of the virus, such as a chemokine monokine induced by IFN-γ (MIG) and IFN-γ-inducible protein 10 (IP10) [69,70]. Therefore, the dry symptom of patients with SS may get worse by SARS-CoV-2 infection. Moreover, SARS-CoV-2 infection may disturb the general immune system, influencing the autoimmune response in SS.

Virus infection essentially changes the expression or regulation of various genes, oncogenes, cell cycle-related genes, cell death-related genes, and immune-regulated genes [71,72,73,74]. Recent in vitro experiments demonstrate that a cell cycle-related protein, retinoblastoma-associated protein 48 (RBAp48), is induced by HIV or HPV infection using [75,76]. RBAp48 controls chromatin organization or assembly to act to induce histone de-acetylation and nucleosome remodeling, thereby inhibiting the process of viral transcription [76,77,78,79]. Therefore, upregulation of RBAp48 is one of the virus-escape mechanisms in human cells. However, previously, the RBAp48 of salivary gland cells in mice and humans was reported to be upregulated by estrogen deficiency to undergo apoptosis [48,80]. Moreover, autoimmune exocrinopathy resembling primary SS was observed in RBAp48-TG mice [48]. Increased RBAp48 expression was also found in the epithelial cells of minor salivary gland of lip biopsy from patients with primary SS [48]. Viral infections, such as HIV and HPV, may change the phenotype or feature of target cells via life or death to trigger a breakdown of local immunological tolerance leading to the onset of autoimmune diseases.

The formation of autoantigen in primary SS is still unclear, although molecular mimicry, bystander activation, and epitope-spreading are immunologically considered as the possible mechanisms. Numerous autoantigens, such as SSA (Ro), SSB (La), muscarinic receptor-3 (M3R), salivary protein 1 (SP1), kallikrein, aquaporin-5 (Aq-5), vasoactive intestinal peptide, α-fodrin, and tripartite motif-38 (TRIM-38), have been reported, based on clinical and animal model studies [81,82]. Before 2000, many researchers focused on the monoclonality of T cells in the pathogenesis of primary SS in which a TCR corresponds to a specific autoantigen. However, as various proteins have been reported as pathogenic autoantigens, it has been argued that multi-clonal T cell responses may induce the autoimmune disorder in primary SS during each step of the autoimmune process. Although the relationship between virus infection and autoantigen formation in primary SS is veiled in mystery, apoptosis of target cells by virus infection can trigger the upregulation of various enzymatic activities to form any pathogenic epitopes from intracellular molecules leading to an autoimmune response.

## 7. Conclusions

SS is a multifactorial disease with a complicated pathogenesis that has so far eluded the establishment of radical therapy. Viral infection mainly interferes with the immune system via autoantigen formation, abnormal functions of various immune cells, or a breakdown of immune tolerance in the target organs. The viral life cycle in the human body is closely related to the breakdown of the immune system and the trigger or progress of autoimmune responses in many autoimmune disorders. Emerging infectious diseases, such as the new coronavirus, are expected to occur time and again to pose such challenges to the human immune system.

## Figures and Tables

**Figure 1 viruses-14-01474-f001:**
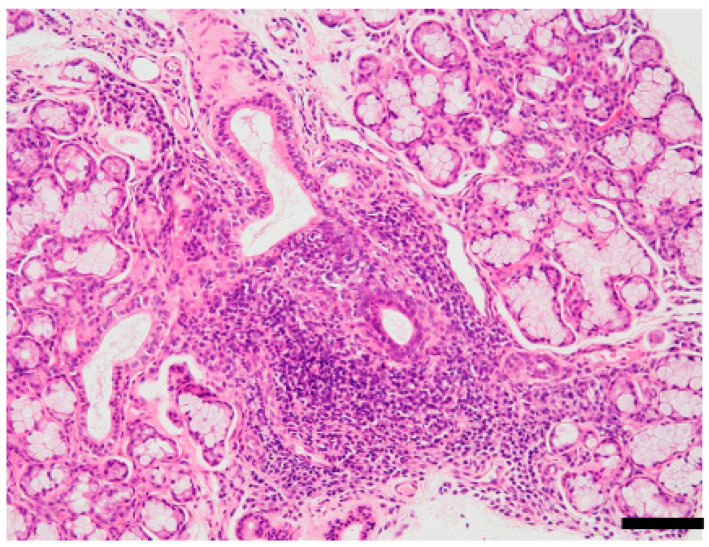
Focal lymphocytic sialadenitis of Sjögren’s syndrome (SS). Hematoxylin and eosin (HE)-stained section of minor salivary glands prepared from lip biopsy from a patient with SS. Notice the lymphocytic infiltration around the salivary ducts and the atrophy of acinar cells. Pathological diagnostic criteria of SS is ≥1 focus score or Greenspan grade ≥ 3 [32,33]. One focus refers to an aggregate of ≥50 mononuclear cells, including lymphocytes, histiocytes, and plasma cells around the ductal structure. Scale bar: 100 μm. The image is unpublished.

**Figure 2 viruses-14-01474-f002:**
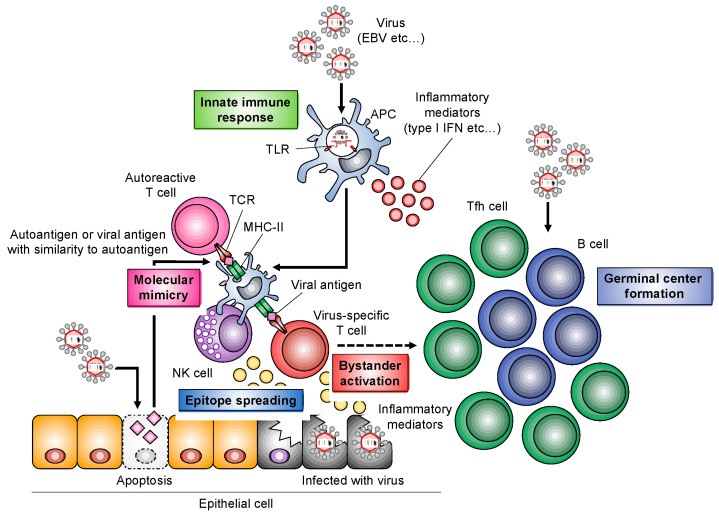
Relationship between virus infection and the pathogenesis of SS. Molecular mimicry, epitope spreading, and bystander activation are among the mechanisms associated with the onset or progress of SS in the target organ or the peripheral immune system.

**Figure 3 viruses-14-01474-f003:**
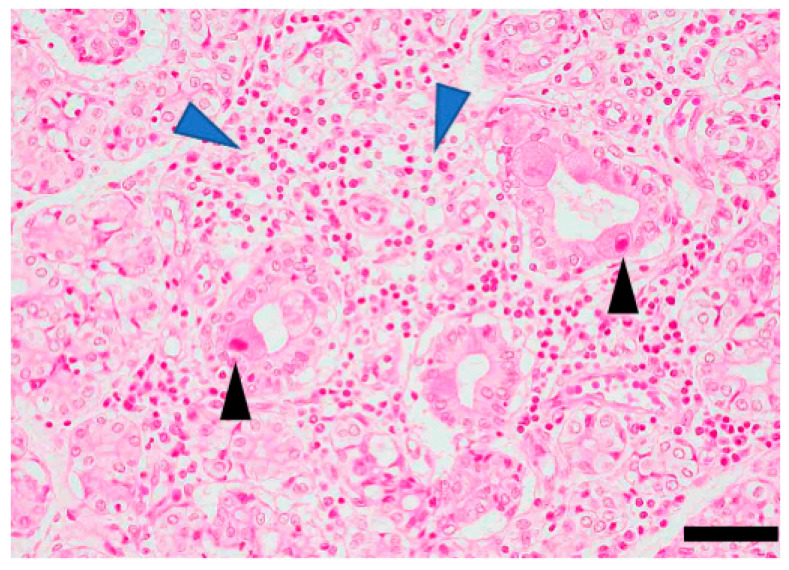
Histopathology of cytomegalovirus (CMV) infection in the salivary gland. HE-stained section of minor salivary glands prepared from lip biopsy. Blue arrowheads indicate inflammatory infiltrates. Black arrowheads indicate intranuclear inclusion bodies. Scale bar: 50 μm. The image is unpublished.

**Figure 4 viruses-14-01474-f004:**
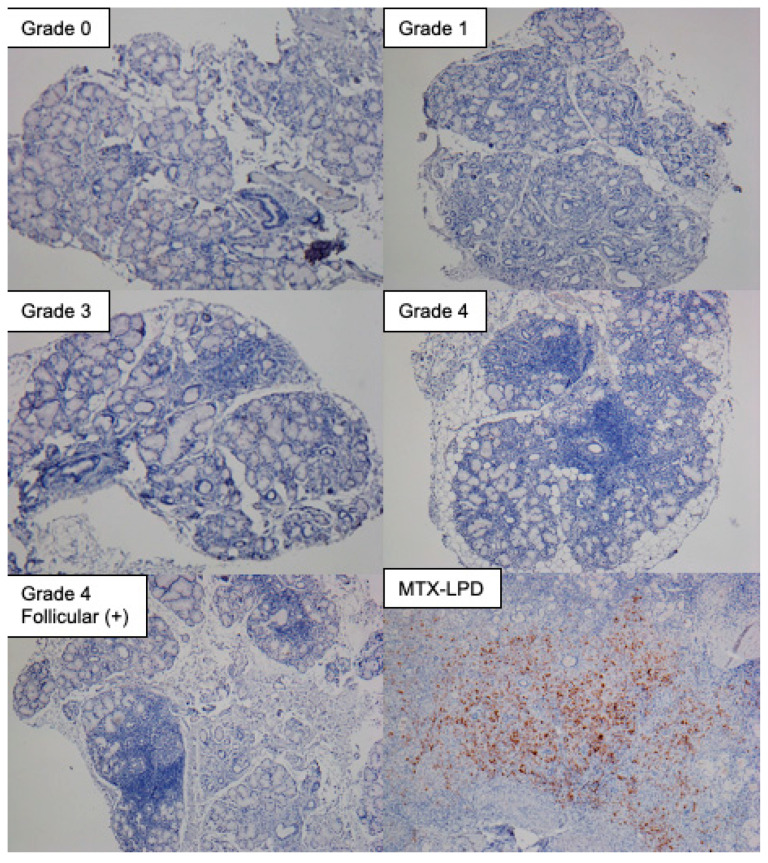
EBNA expression in minor salivary gland tissues from SS and MTX-LPD patients. Samples graded 0 to 4 according to Greenspan grade [40,41] were used for this analysis. Grade 0 = the absence of lymphocytes and plasma cells per 4 mm^2^ in labial salivary glands (LSGs), Grade 1 = mild infiltration of lymphocytes and plasma cells per 4 mm^2^ in LSGs, Grade 2 = a moderate infiltration or less than one focus per 4 mm^2^ in LSGs, Grade 3 = a single focus per 4 mm^2^ in LSGs, and Grade 4 = more than one focus per 4 mm^2^ in LSGs. The stained sample shown was from a gingival mucosa biopsy in a patient with MTX-LPD. Immunohistochemical analysis, performed using paraffin-embedded sections, used FLEX monoclonal mouse anti-EBV, LMP antibody (Clone: CS.1–4, DAKO) as the primary antibody, and SignalStain Boost IHC (horseradish peroxidase: HRP, mouse) (Cell Signaling) as the secondary antibody. EBNA expression was visualized by 3,3-diaminobensidine (DAB) reaction, and counterstaining was performed by hematoxylin. The image is unpublished.

## Data Availability

The data presented in this study are available in the included articles. No new data were created or analyzed in this study.

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
