# Peer review of "Virus Infections Play Crucial Roles in the Pathogenesis of Sjögren’s Syndrome"

_viruses, 2022, doi:10.3390/v14071474_

Round 1
Reviewer 1 Report
The manuscript ‘Crucial Roles of Virus Infection in the Pathogenesis of Sjögren’s Syndrome’ by Otsuka and colleagues has been reviewed.
- The topic has already been covered by many comprehensive reviews published earlier.
- The manuscript is too short and too little information is provided about about the relationship between viral infections and SS, and about the findings of recent research.
- Since this work is submitted to a reputable journal (Viruses), the authors must improve the manuscript making it much more comprehensive by covering the latest significant research data to the existing field of research. I also would recommend focusing on specific aspects and writing a new section as future directions/perspectives.
- I suggest to change the manuscript title to:
Viral Infections Play Crucial Roles in the Pathogenesis of Sjögren’s Syndrome.
- Sections 3, 4 and 5 have the same title!! Please revise and be specific.
- From the title "Crucial Roles of Virus Infection in the Pathogenesis......", I expected more details with molecular and cellular mechanisms (with original figures) elucidating the relationship between SS and the infection with EBV, HTLV, HSV, HBV, HCV, and CMV. However, the authors discussed this important information in very brief paragraphs.
- It is worth to discuss the the potential roles of each virus in the pathogenesis of SS in an independent section with enough information. In addition to the basic pathophysiology information and molecular and cellular mechanisms (original figures are recommended), each section should include an up-to-date review of recent research studies.
- The authors should think on how to highlight the novelty of this work? how it would be of interest to readers? and why readers may prioritize this work from the previously published reviews?
- Coherence, balance and writing flow are essential keys to produce a good and interesting review article; writing a review article is not only gathering information about a specific topic from the relevant references.
- Embed each figure in the order it first mentioned in the text. Are these figures (1 - 4) original? If no, did the authors get copyright permissions?
Author Response
- The topic has already been covered by many comprehensive reviews published earlier.
Answer: We have added the more detailed information and discussion in the revised manuscript.
- The manuscript is too short and too little information is provided about the relationship between viral infections and SS, and about the findings of recent research.
Answer: As suggested by reviewer, the firstly submitted manuscript is too compact. We have added more detailed description in the revised manuscript.
- Since this work is submitted to a reputable journal (Viruses), the authors must improve the manuscript making it much more comprehensive by covering the latest significant research data to the existing field of research. I also would recommend focusing on specific aspects and writing a new section as future directions/perspectives.
Answer: According to the comment by reviewer, we have added a new section “Future Perspectives” in the revised manuscript.
- I suggest to change the manuscript title to:
Viral Infections Play Crucial Roles in the Pathogenesis of Sjögren’s Syndrome.
Answer: According to the comment by reviewer, we have changed the title.
- Sections 3, 4 and 5 have the same title!! Please revise and be specific.
Answer: The titles have been modified in the revised manuscript.
- From the title "Crucial Roles of Virus Infection in the Pathogenesis......", I expected more details with molecular and cellular mechanisms (with original figures) elucidating the relationship between SS and the infection with EBV, HTLV, HSV, HBV, HCV, and CMV. However, the authors discussed this important information in very brief paragraphs.
Answer: We have added the more detailed information and discussion in the revised manuscript.
- It is worth to discuss the potential roles of each virus in the pathogenesis of SS in an independent section with enough information. In addition to the basic pathophysiology information and molecular and cellular mechanisms (original figures are recommended), each section should include an up-to-date review of recent research studies.
Answer: We have added the more detailed information and discussion in the revised manuscript. In particular, the description as to molecular mechanisms using animal models has been added to the revised manuscript.
- The authors should think on how to highlight the novelty of this work? how it would be of interest to readers? and why readers may prioritize this work from the previously published reviews?
Answer: We have added the more detailed information and discussion in the revised manuscript.
- Coherence, balance and writing flow are essential keys to produce a good and interesting review article; writing a review article is not only gathering information about a specific topic from the relevant references.
Answer: We have focused on more specific topic, and added the more detailed information and discussion in the revised manuscript.
- Embed each figure in the order it first mentioned in the text. Are these figures (1 - 4) original? If no, did the authors get copyright permissions?
Answer: All figures are original.

Reviewer 2 Report
The authors describe extensively the role of viruses in the pathogenesis of the disease.
However, a more detailed comparison of the role of individual viruses in Sjogren's syndrome would be desirable.
There are inaccuracies: incorrect definition of primary Sjogren's syndrome (lines 25-29). The titles of subsections 3, 4, 5 are the same. There are spelling mistakes.
Author Response
However, a more detailed comparison of the role of individual viruses in Sjogren's syndrome would be desirable.
There are inaccuracies: incorrect definition of primary Sjogren's syndrome (lines 25-29). The titles of subsections 3, 4, 5 are the same. There are spelling mistakes.
Answer: The titles have been modified in the revised manuscript.

Reviewer 3 Report
In this Review manuscript, Dr. Kunihiro Otsuka and colleagues present a comprehensive overview that delves into the possible/potential role of viruses in the development and/or onset of Sjögren´s Syndrome (SS) in the human population. This topic has been discussed for decades, especially the posible role of EBV as an initiator of SS, but the subject remains without a definitive conclusión. For this Reviewer, a main reason, and one that plagues this entire field, is the fact that the viruses chosen as candidates to study are highly prevalent in the general human population wherethe vast majority of individuals are not positive for actually developing the autoimmune SS disease. Thus, the appearance or presence of antibodies toward these viruses in individuals diagnosed with SS remains questionable for being an initiator of SS pathology, even if one tries to invoke genetic susceptibility. Thus, until immunization of susceptible individuals with a specific virus actually triggers autoimmune SS disease, one can only suggest an association, as the authors have pointed out (lines 80-83). How this fact influences the topic of viruses in SS is not adequately discussed.
While the authors point out that the viruses in question are not only highly prevalent in normal human populations, they do not discuss the topic from a more molecular aspect. The authors have acknowledged that the basic literature indicates that the TLRs involved in human SS and mouse SS are basically TLR7 and TLR3, respectively, suggesting the initial glandular responses are against ssRNA and dsRNA, respectively. However, viruses being discussed here are: CMV, EBV, HSV, and HBV which are dsDNA viruses, HTLV a sRNA virus, and HCV the only ssRNA virus. One might rule out all of thes mentioned viruses in the mouse SS pathology since none are dsRNA viruses, but dsRNA and sRNA viruses are a fascinating viral entity also known to be associated with autoimmunity.
A second set of data presented by the authors suggests that the ¨inflammatory lesions in the target organ are formed ….. via the breakdown of central or peripheral immunological tolerance, induced by genetic and/or environmental factors, including viral infection¨. One could argue that the premise of a breakdown in immunological tolerance is not correct, as this could be a natural response to viral antigens that just happen to crossreact with self antigenes at the molecular level (e.g., antigenic mimicry). Afterall, autoantibodies to glandular tissue are present in SS prior to any LF formation and this could represent an attempt by the organ to protect itself in a normal individual. How would this fact factor into the overall concept when one considers genetics?
Another interesting statement made by the authors (line 186-190) is ¨ increased expression levels of type 1 IFN or type 1 IFN-induced genes have been observed in a lot of patients with SLE and other autoimmune diseases¨. This statement suggests that the Interferon signatures for SS have yet to be determined; however, IFN signatures have been published for both human and mouse SS pathology more than a decade.
Lastly, there is little discussion of the wide disparities in sicca syndromes, i.e., general dryness of eyes and mouth with and without autoimmunity. Would this alter the discussion point since viruses could easily cause dryness in the absence of a fully diagnosed SS pathology, e.g., aging?
Author Response
To Reviewer #3,
In this Review manuscript, Dr. Kunihiro Otsuka and colleagues present a comprehensive overview that delves into the possible/potential role of viruses in the development and/or onset of Sjögren´s Syndrome (SS) in the human population. This topic has been discussed for decades, especially the possible role of EBV as an initiator of SS, but the subject remains without a definitive conclusión. For this Reviewer, a main reason, and one that plagues this entire field, is the fact that the viruses chosen as candidates to study are highly prevalent in the general human population where the vast majority of individuals are not positive for actually developing the autoimmune SS disease. Thus, the appearance or presence of antibodies toward these viruses in individuals diagnosed with SS remains questionable for being an initiator of SS pathology, even if one tries to invoke genetic susceptibility. Thus, until immunization of susceptible individuals with a specific virus actually triggers autoimmune SS disease, one can only suggest an association, as the authors have pointed out (lines 80-83). How this fact influences the topic of viruses in SS is not adequately discussed.
Answer: As pointed out by reviewer, the mechanism of autoantigen formation in SS has been still unclear. We have added the information and discussion in the Future perspectives section of the revised manuscript.
While the authors point out that the viruses in question are not only highly prevalent in normal human populations, they do not discuss the topic from a more molecular aspect. The authors have acknowledged that the basic literature indicates that the TLRs involved in human SS and mouse SS are basically TLR7 and TLR3, respectively, suggesting the initial glandular responses are against ssRNA and dsRNA, respectively. However, viruses being discussed here are: CMV, EBV, HSV, and HBV which are dsDNA viruses, HTLV a sRNA virus, and HCV the only ssRNA virus. One might rule out all of thes mentioned viruses in the mouse SS pathology since none are dsRNA viruses, but dsRNA and sRNA viruses are a fascinating viral entity also known to be associated with autoimmunity.
Answer: According to the comment by reviewer, the description has modified not to lead to misunderstand in the revised manuscript.
A second set of data presented by the authors suggests that the ¨inflammatory lesions in the target organ are formed ….. via the breakdown of central or peripheral immunological tolerance, induced by genetic and/or environmental factors, including viral infection¨. One could argue that the premise of a breakdown in immunological tolerance is not correct, as this could be a natural response to viral antigens that just happen to crossreact with self antigenes at the molecular level (e.g., antigenic mimicry). Afterall, autoantibodies to glandular tissue are present in SS prior to any LF formation and this could represent an attempt by the organ to protect itself in a normal individual. How would this fact factor into the overall concept when one considers genetics?
Answer: As pointed out by reviewer, the description may lead to misunderstand. We have modified the description in the revised manuscript.
Another interesting statement made by the authors (line 186-190) is ¨ increased expression levels of type 1 IFN or type 1 IFN-induced genes have been observed in a lot of patients with SLE and other autoimmune diseases¨. This statement suggests that the Interferon signatures for SS have yet to be determined; however, IFN signatures have been published for both human and mouse SS pathology more than a decade.
Answer: As pointed out by reviewer, IFN signatures have already been reported in SS patients. We modified the description in the revised manuscript.
Lastly, there is little discussion of the wide disparities in sicca syndromes, i.e., general dryness of eyes and mouth with and without autoimmunity. Would this alter the discussion point since viruses could easily cause dryness in the absence of a fully diagnosed SS pathology, e.g., aging?
Answer: We have focused on more specific topic, and added the more detailed information and discussion in the revised manuscript.
Reviewer 4 Report
This is an up-to-date as well as concise review on the roles of viruses on the pathogenesis of Sjögren’s syndrome (SS). However, in the current version, there are several points which should be amended before publication.
The authors are kindly asked to check the points below.
1. Chapter title: Chapter 3-5 have the same title ‘Sjögren’s syndrome and viral infection’. Please give an adequate title for each chapter.
2. The description of Chapter 3, line 28 to 30 (A recent study…(SARS-CoV-2) infects…) is a repetition of Chapter 1, line 27 to 29. In my opinion, that of Chapter 1 is not necessary.
3. Chapter 4, line 3: ‘abave’ should be ‘above’.
4. Chapter 5, line 26: it is not clear what ‘SS-Bm’ means.
5. Are Figure 1 and Figure 3 the authors’ own works or cited by the references from other labs? Please denote the images are the authors’ unpublished work or already-published work. If they are already-published images, please give reference titles for each figure.
6. Figure 3 are explained to show intranuclear inclusion body and lymphocytic infiltration in CMV-infected salivary glands in the main document, the description of which does not correspond to figure legend (merely mentions intranuclear inclusion body). Please add an adequate sign to indicate the site of lymphocytic infiltration on Figure 3.
7. Title of legend for Figure 4 says the images are from SS patients. Is the panel ‘MTX-LPD’ from a SS patient?
8. Keywords: it is not clear what ‘S Virus’ means.
Author Response
To Reviewer #4,
This is an up-to-date as well as concise review on the roles of viruses on the pathogenesis of Sjögren’s syndrome (SS). However, in the current version, there are several points which should be amended before publication.
The authors are kindly asked to check the points below.
- Chapter title: Chapter 3-5 have the same title ‘Sjögren’s syndrome and viral infection’. Please give an adequate title for each chapter.
Answer: The titles have been modified in the revised manuscript.
- The description of Chapter 3, line 28 to 30 (A recent study…(SARS-CoV-2) infects…) is a repetition of Chapter 1, line 27 to 29. In my opinion, that of Chapter 1 is not necessary.
Answer: According to the comment by reviewer, we deleted the description in the revised manuscript.
- Chapter 4, line 3: ‘abave’ should be ‘above’.
Answer: We have modified the spelling in the revised manuscript.
- Chapter 5, line 26: it is not clear what ‘SS-Bm’ means.
Answer: The spelling has been corrected in the revised manuscript.
- Are Figure 1 and Figure 3 the authors’ own works or cited by the references from other labs? Please denote the images are the authors’ unpublished work or already-published work. If they are already-published images, please give reference titles for each figure.
Answer: The Figure 1 and 3 are original. The information has been added in the Figure legend section of the revised manuscript.

Round 2
Reviewer 1 Report
The paper can be accepted for publication.